# Automatic and accurate reconstruction of long-range axonal projections of single-neuron in mouse brain

Lin Cai[1,2], Taiyu Fan[1,2], Xuzhong Qu[1,2], Ying Zhang[1,2], Xianyu Gou[1,2], Quanwei Ding[1,2,3], Weihua Feng[1,2], Tingting Cao[1,2], Xiaohua Lv[1,2], Xiuli Liu[1,2], Qing Huang[1,2,3], Tingwei Quan[1,2]*, Shaoqun Zeng[1,2]

[1]Britton Chance Center for Biomedical Photonics, Wuhan National Laboratory for Optoelectronics, Huazhong University of Science and Technology, Wuhan, China; [2]MOE Key Laboratory for Biomedical Photonics, Wuhan National Laboratory for Optoelectronics, Huazhong University of Science and Technology, Wuhan, China; [3]School of Computer Science and Engineering, Hubei Key Laboratory of Intelligent Robot, Wuhan Institute of Technology, Wuhan, China

*For correspondence: quantingwei@hust.edu.cn

Competing interest: The authors declare that no competing interests exist.

## eLife Assessment

This **important** paper takes a novel approach to the problem of automatically reconstructing long-range axonal projections from stacks of images. The key innovation is to separate the identification of sections of an axon from the statistical rules used to constrain global structure. The authors provide **compelling** evidence that their method is a significant improvement over existing measures in circumstances where the labelling of axons and dendrites is relatively dense.

**Abstract** Single-neuron axonal projections reveal the route map of neuron output and provide a key cue for understanding how information flows across the brain. Reconstruction of single-neuron axonal projections requires intensive manual operations in tens of terabytes of brain imaging data and is highly time-consuming and labor-intensive. The main issue lies in the need for precise reconstruction algorithms to avoid reconstruction errors, yet current methods struggle with densely distributed axons, focusing mainly on skeleton extraction. To overcome this, we introduce a point assignment-based method that uses cylindrical point sets to accurately represent axons and a minimal information flow tree model to suppress the snowball effect of reconstruction errors. Our method successfully reconstructs single-neuron axonal projections across hundreds of GBs (Gigabytes) images within a mouse brain with an average of 80% f1-score, while current methods only provide less than 40% f1-score reconstructions from a few hundred MBs (Megabytes) images. This huge improvement is helpful for high-throughput mapping of neuron projections.

## Introduction

Neuronal axons in general project to different brain regions, and their projection distribution is an essential cue for neuron type identification, neuronal circuit construction, and deeper insight into how information flows in the brain (*Huang and Luo, 2015*; *Meijering, 2010*; *Parekh and Ascoli, 2013*; *Zingg et al., 2014*). Advances in optical imaging and molecular labeling techniques (*Cai et al., 2019*; *Chung and Deisseroth, 2013*; *Çiçek et al., 2016*; *Kim and Schnitzer, 2022*; *Li et al., 2010*; *Osten and Margrie, 2013*) have allowed us to observe the entire mouse brain at single-axon resolution and provided the database for the study of neuronal projection patterns (*Foster et al., 2021*; *Gao et al.,*

*2022*; *Muñoz-Castañeda et al., 2021*; *Peng et al., 2021*; *Qiu et al., 2024*; *Sun et al., 2019*; *Xu et al., 2021*; *Zeng, 2022*). However, the reconstruction of these long-range projected axons still requires extensive manual annotation in tens of TBs volumetric images (*Çiçek et al., 2016*; *Friedmann et al., 2020*; *Wang et al., 2019*; *Winnubst et al., 2019*; *Zhou et al., 2021*), this labor-intensive process creates a major bottleneck for high-throughput mapping of neuronal projections (*Zeng and Sanes, 2017*).

The difficulties in reconstructing the long-range projections of neurons are as follows. On the one hand, while molecular labeling techniques can shed light on a very small fraction of neurons, a significant fraction of neuronal axons is still densely distributed due to the morphological complexity of neurons. The identification of densely distributed axons is considered an open problem in the field (*Li et al., 2019*; *Lichtman and Denk, 2011*; *Zeng and Sanes, 2017*), which still has no good solution. On the other hand, during neuron reconstruction, reconstruction errors accumulate, and a single reconstruction error can result in an entire branch being connected erroneously to other neurons or missing (*Helmstaedter, 2013*). Therefore, effective large-scale reconstruction of neurons requires extremely high identification accuracy of dense axons. The contradictions between these two aspects seem hard to reconcile.

The current neuron reconstruction frameworks focus on how to accurately extract skeletons of neurites and establish the connections between skeletons (*Meijering, 2010*; *Peng et al., 2015*). The BigNeuron project (*Manubens-Gil et al., 2023*) conducts a systematic evaluation of 35 automatic neuron reconstruction algorithms, all of which are based on tracing neurite skeletons and can be divided into two categories: local and global approaches. In the local approach (*Choromanska et al., 2012*; *Li et al., 2020*; *Peng et al., 2011*; *Yang et al., 2013*), the localization of the next skeleton point requires computation of the signal anisotropy of the image region near the current skeleton point. Localization errors typically occur when this image region contains other neurite signals. The global approach (*Li et al., 2019*; *Türetken et al., 2011*; *Xiao and Peng, 2013*) first generates multiple seed points that are commonly located at the neurite centerline and then establishes connections between these seed points for generating the neurite skeleton. This connection relies mainly on spatial location information, resulting in densely distributed neurites being connected to each other erroneously. While deep learning is widely used in neuron reconstruction (*Huang et al., 2020*; *Li and Shen, 2020*; *Liu et al., 2022*; *Zhou et al., 2018*), - mainly for neuronal image segmentation and signal intensity enhancement to reduce reconstruction errors - even ideal segmentation with all neurite centers identified and their signal enhanced still exhibits significant reconstruction errors with skeleton-based methods (*Figure 1—figure supplement 1*).

To address the problem of error accumulation during neuron reconstruction, it is common practice to utilize statistical information of neuron morphology, such as the angle between two neurites, to identify and remove spurious connections between the reconstructed neurites. This strategy (*Li et al., 2019*; *Quan et al., 2016*) achieves 80% reconstruction accuracy from GB-scale images under two critical constraints: (1) precise identification of neurite terminals and branch points is required for accurate angle computation and morphological analysis, and (2) somatic locations are required as critical information to remove some links between the reconstructed neurites to ensure that each cell body can be mapped to the root node of a single tree structure. However, for long-range axonal reconstruction across hundreds of GB-scale images, the strategy is not effective to eliminate the accumulation of errors due to factors such as the position of the axon at a distance from the soma and slight morphological differences between axon junction and termination. Consequently, current long-range projection reconstruction methods are semi-automatic and require substantial human intervention (*Gao et al., 2023*; *Wang et al., 2019*; *Winnubst et al., 2019*; *Zhou et al., 2021*).

Here, we propose a new neuron reconstruction method called PointTree, which aims at how to assign foreground points in neuronal images to their own neurons. In the workflow, we design a constrained Gaussian clustering method to partition the foreground region of a neuronal image into a series of columnar regions whose centerline belongs to only a single neurite. This operation essentially eliminates the interference of different neurites in the dense reconstruction. In addition, each columnar region is characterized by a minimal envelope ellipsoid for constructing connections between columnar regions, which forms the neurite shapes. Based on the reconstructed shapes, we design a minimal information flow tree model to suppress the cumulative reconstruction error. Using

the proposed method, we successfully achieve accurate reconstruction of long-range projections of neurons across hundreds of gigabytes of volumetric image.

## Results

### The architecture and principles of PointTree

In the design of PointTree, we have developed a series of optimization problems to assign foreground points in data blocks to their respective neurites. Firstly, the segment network is utilized for each data block to obtain foreground points. Subsequently, we apply a constrained Gaussian clustering method (*Reynolds, 2009*) to partition the foreground points into columnar regions and determine their geometrical parameters by solving the minimum-volume covering ellipsoids problem (*Sun and Freund, 2004*). Using these geometrical parameters, we construct a 0–1 assignment problem (*Volgenant, 1996*) to establish links between these columnar regions. Finally, skeletons are extracted from these linked columnar regions to reduce data redundancy by using region growing (*Harris, 2011*). The key procedures for neuron reconstruction are presented in *Figure 1A*.

In addition, PointTree employed the statistical prior information to reduce the reconstruction errors. At the branching point (node) of the neurites, it can be divided into three segments of neurite skeletons. The segment entering the node forms two angles with the other two segments exiting the node respectively. The node angle is defined as the smaller angle between the entering segment and each exiting segment (*Figure 1B*). With node angle, we can identify the single complete neurite and its corresponding node angles. The skeleton of the neurite is generally smooth, with very few sudden directional changes and even fewer at the nodes. So, the node angles should be as small as possible. For neuronal branches, the node angles are uniquely determined when the root node is given, and the sum of the negative cosine of these node angles expressed by information flow value is small when the root node is correctly identified. This rule is defined as a minimal information flow tree (MIFT).

In image blocks of densely distributed neurites, we used semi-automatic software (*Zhou et al., 2021*) extracting 208 neuronal branches and identifying their root nodes. For each branch, we calculated their information flow values as the ground-truth information flow values (*Figure 1C*). To validate MIFT, we looped through all possible structures of these branches by changing the root node in order to compute the maximum and minimum information flow values (*Figure 1C*). It is evident that, for most neuronal branches (195/208), the ground-truth values of the information flow achieve the minimum value, suggesting that MIFT rule is reasonable. We utilized MIFT to modify skeleton structure and remove spurious connections between reconstructed neurites (*Figure 1D* and *Figure 1—figure supplement 2*), both for reconstructions within individual blocks and for the fused reconstruction in adjacent blocks.

PointTree has the capability to separate densely distributed neurites. When dealing with two parallel neurites in close proximity to each other, their shapes can be represented by a series of columnar regions (the left panels of *Figure 1E*). We have modified the Gaussian clustering algorithm by constraining the estimated mean and covariance parameters so that the cluster shape approaches a columnar shape. Additionally, foreground points within the same cluster are connected to each other. These two features ensure that the central line in the columnar region belongs to only a single neurite, which is crucial for separating densely packed neurites. Furthermore, we utilize the minimum volume covering ellipsoid to extract shape information of the columnar regions for constructing their connections. These designs enable PointTree to successfully reconstruct packed neurites. In contrast, skeleton-based local methods rely on determining the position of the next skeleton point based on the shape anisotropy of the region. This often leads to localization errors when there are two neurite image signals within a region (the middle panels of *Figure 1E*). When it comes to skeleton-based global methods, although seed points can be located at individual neurite centers, accurately constructing connections between these seed points proves challenging due to the reliance on distance between points and susceptibility to interference from densely distributed neurites (the right panels of *Figure 1E*).

### The merits of PointTree in dense reconstruction

In dense reconstruction, one of the main concerns is how well to separate densely distributed neurites that behave as crossover and closely paralleled neurites. These neurites can be manually identified

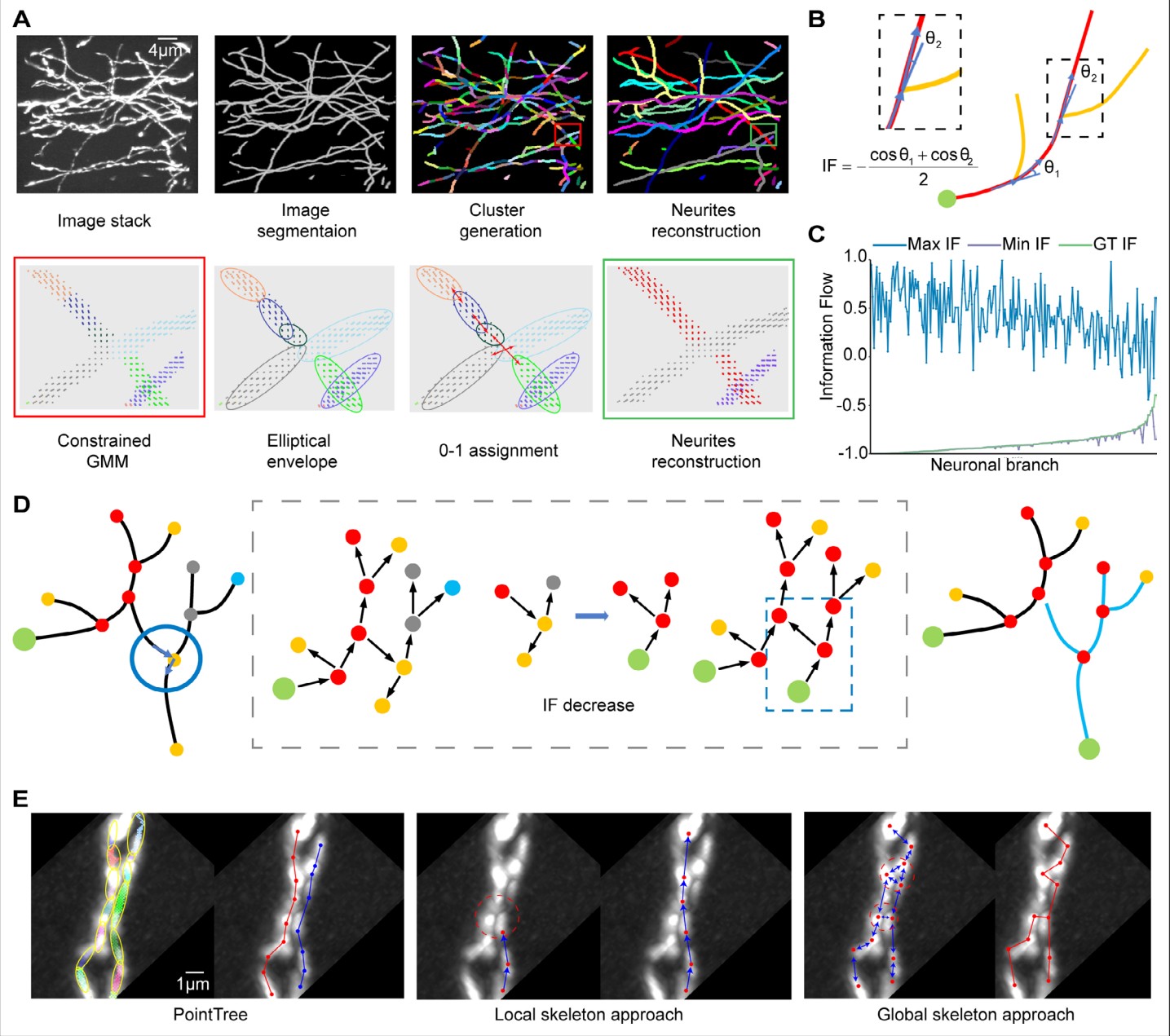

**Figure 1.** Summary and principle of PointTree. (**A**) The reconstruction procedure of PointTree involves the generation, clustering, and connection of foreground points (the first row). Within this procedure, three optimization problems are designed to allocate the foreground points into their respective neurites (the second row). (**B**) Schematic diagram of information flow score calculation. In a neurite branch with a fixed root node (green circle), the information flow score is calculated based on the assumption that a neurite has few directional changes. The assumption determines the neurite directly connecting to the root node (red), resulting in two branch angles used to calculate the information flow score. (**C**) Statistical analysis of the consistency between the minimum information flow and the real situation. For 208 neurite branches, the information flow scores are calculated as ground truth according to their manually determined skeletons and root nodes. These scores are then displayed in ascending order. The root nodes of neurite branches are changed to generate both maximum and minimum information flow scores. (**D**) One neurite branch is decomposed into two by minimizing the total information flow scores. (**E**) Performance of different methods on separating closely paralleled neurites. In PointTree, a single neurite is represented by a series of ellipsoids whose centerlines are not simultaneously located within different neurites. They are connected using an ellipsoid shape, which results in perfect reconstruction (Left). However, skeleton-based methods fail to separate two closely paralleled neurites due to interference from other signals (Red circle in middle) or connections being interfered with by another neighboring skeleton point (Red circle in right).

The online version of this article includes the following figure supplement(s) for figure 1:

**Figure supplement 1.** Comparison of PointTree and several skeleton-based methods for reconstructing the segmented image block derived from ground-truth skeletons.

*Figure 1 continued on next page*

*Figure 1 continued*

**Figure supplement 2.** The generation of minimal information flow tree.

**Figure supplement 3.** Post-processing for structures violating the MIFT rule.

**Figure supplement 4.** Tree structure derived from SWC file that records the axonal reconstruction.

by visualization with different view angles (*Figure 2—figure supplement 1*). We compared PointTree with several skeleton-based methods such as neuTube (*Feng et al., 2015*), PHDF (*Radojevic and Meijering, 2017*), NGPST (*Quan et al., 2016*), and MOST (*Wu et al., 2014*) in performing this task. We manually labeled the locations where neurites are crossover or closely parallel from five 256×256 × 256 image blocks. For a fair comparison, all methods are performed on segmented images derived from the segmentation network. *Figure 2A* illustrates the process of PointTree's separation of crossover and closely paralleled neurites. PointTree can successfully separate the densely distributed neurites in a range of 71.4% and 91.7%, while these skeleton-based methods only separate 25.0% densely distributed neurites (*Figure 2B*) at most. We also present the comparison of PointTree and other methods on some reconstruction examples in which multi-crossover neurites (*Figure 2C*) and closely paralleled neurites are involved. PointTree provides the perfect reconstruction while other methods fail to reconstruct these neurites.

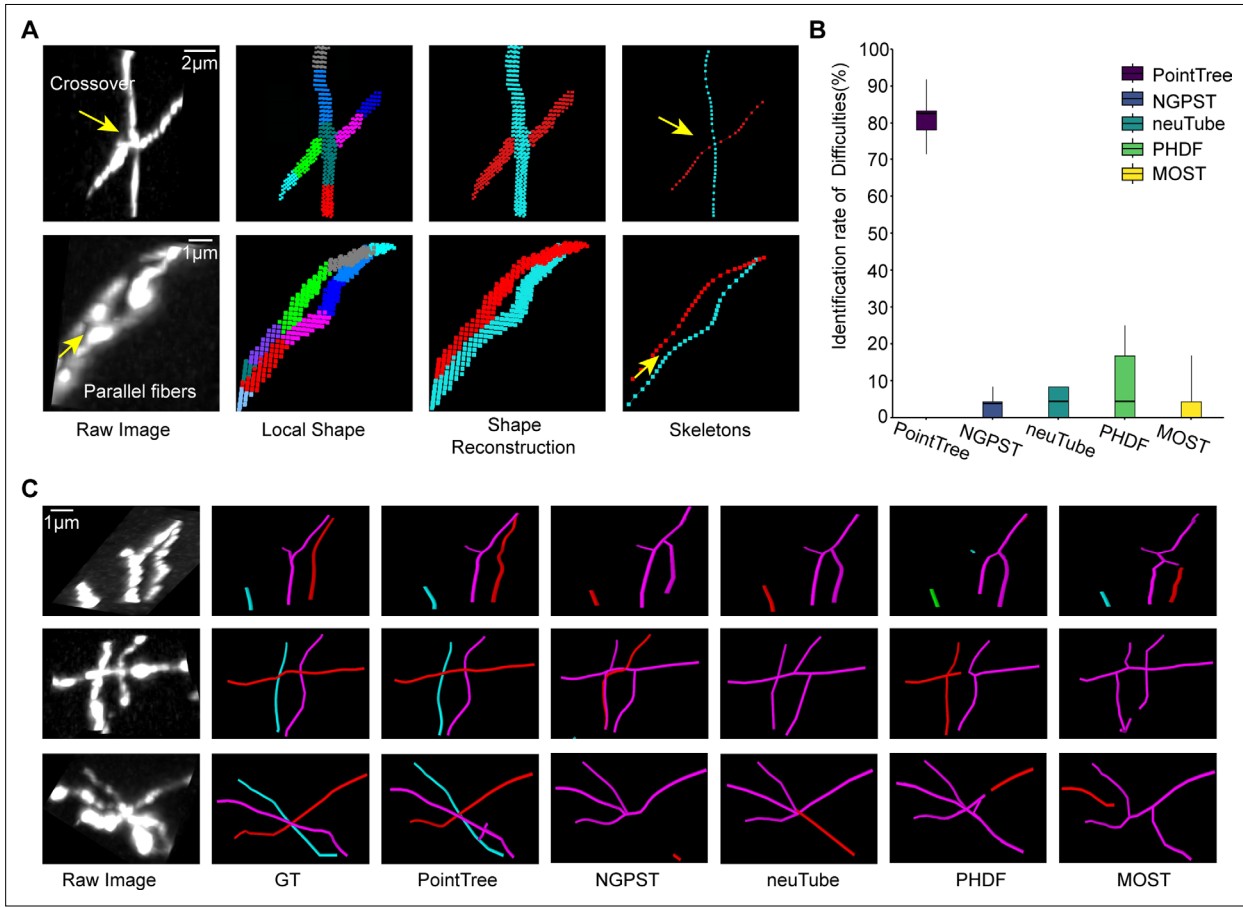

**Figure 2.** Performance of PointTree on crossover and closely paralleled neurites. (**A**) The reconstruction process of crossover and closely paralleled neurites. (**B**) Quantitative evaluation of PointTree and several skeleton-based methods on identifying closely distributed neurites. The box plots present the statistical information (n=5) in which the horizontal line in the box, the lower and upper borders of the box represent the median value, the first quartile (**Q1**), and the third quartile (**Q3**), respectively. The vertical black lines indicate 1.5 × IQR. (**C**) Three reconstruction examples derived from PointTree and several skeleton-based methods.

The online version of this article includes the following figure supplement(s) for figure 2:

**Figure supplement 1.** Visualization of two parallel neurites with different viewing angles.

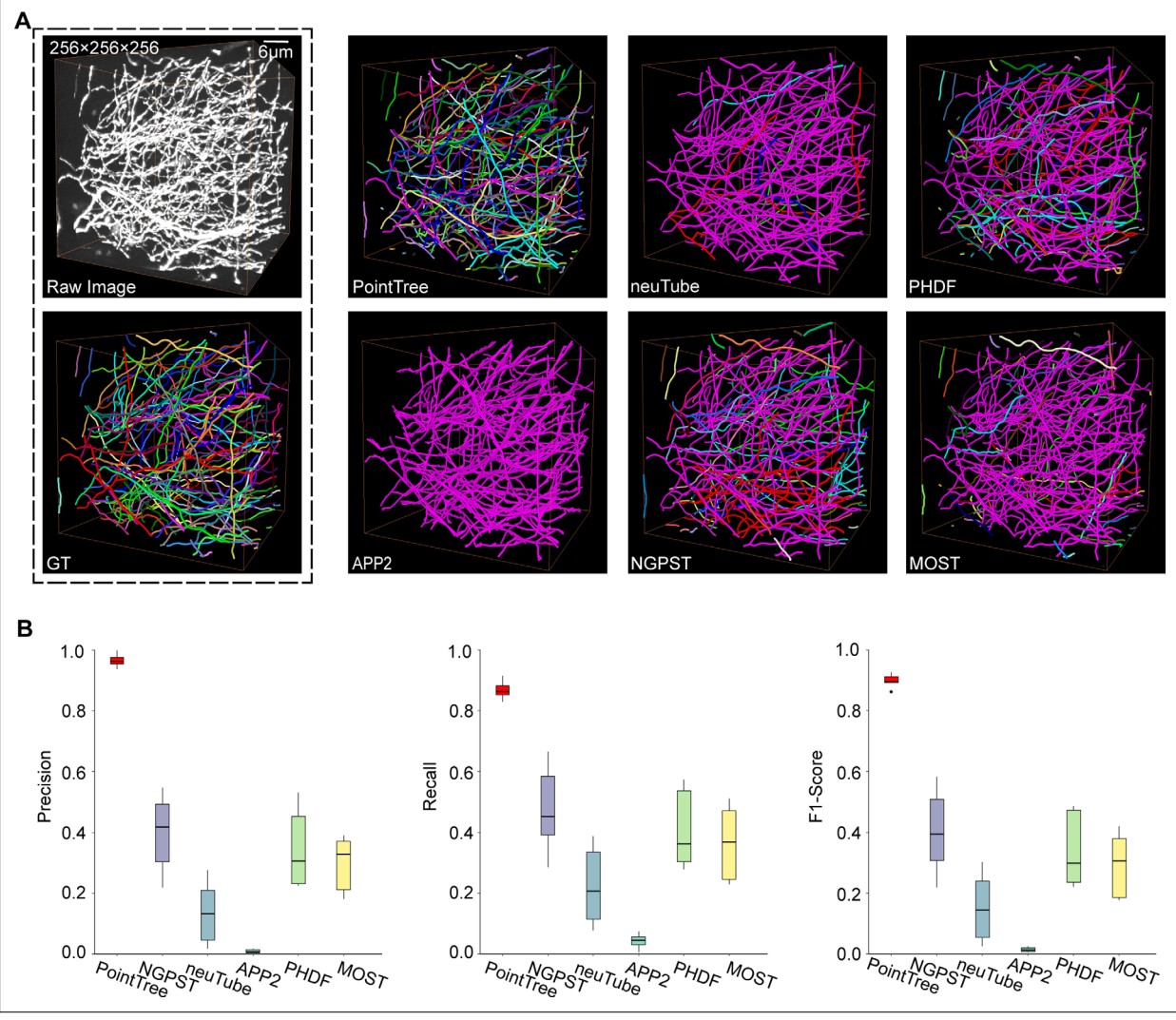

**Figure 3.** Comparison of reconstruction methods on image blocks containing densely distributed neurites. (**A**) Comparison of reconstruction performance among six methods, including PointTree, NGPST, neuTube, APP2, PHDF, and MOST. Individual neurite branches are delineated in different colors. (**B**) Quantitative evaluation of reconstruction performance using precision, recall, and f1-score. The box plots display these three evaluation indexes (n=8). In the box, the horizontal line represents the median value. The box shows the interquartile range (IQR) from the first quartile (**Q1**) to the third quartile (**Q3**). The vertical lines indicate 1.5×IQR.

Furthermore, we present the quantitative results derived from PointTree and five widely used skeleton-based reconstruction methods, including APP2, neuTube, NGPST, PHDF, and MOST. Eight 256×256 × 256 image blocks that include many densely distributed neurites are of the testing dataset. All reconstruction algorithms are performed on the segmentation images of these testing datasets. We give the intuitive reconstruction comparisons (*Figure 3A*). PointTree provides the reconstruction close to the ground truth. The skeleton-based methods generate lots of reconstruction errors and incorrectly combine multi-neurites into a single branch. The quantitative reconstructions suggest that PointTree is far superior to skeleton-based methods (*Figure 3B*). For PointTree, the average precision is above 90%, both recall and f1-score are above 85%. The skeleton-based methods cannot provide a good solution to separate the densely packed neurites. The f1-score of these reconstructions ranges from 30% to 40%, which indicates the ineffective reconstructions.

## Reconstruction of data with different signal-to-noise ratios

In the field of neuronal reconstruction, data acquired by different imaging systems often exhibit varying signal-to-noise ratio (SNR) characteristics. For some low-SNR datasets, severe noise interference

makes it difficult even for human observers to accurately identify neurite structures. To systematically evaluate PointTree's reconstruction performance across datasets with different SNRs, we selected and analyzed data from three imaging systems: light sheet microscopy (*Stelzer et al., 2021*) (LSM), fluorescent micro-optical sectioning tomography (*Wang et al., 2021*) (fMOST), and high-definition fluorescent micro-optical sectioning tomography (*Zhong et al., 2021*) (HD-fMOST), with SNR ranges of 2–7, 6–12, and 9–14, respectively (*Figure 4A*).

Experimental results demonstrate that, thanks to the powerful feature extraction capability of the deep learning network, the trained neural network achieves satisfactory segmentation performance (third row in *Figure 4B*) even on low-SNR data (first two columns in *Figure 4B*, top row), laying a solid foundation for subsequent accurate reconstruction (bottom row in *Figure 4B*). Quantitative analysis reveals that PointTree delivers stable reconstruction performance across all SNR levels. Specifically: for LSM data (sample size n=25, mean SNR = 5.01), average precision = 96.0%, recall = 88.7%, and f1-score=91.0%; for fMOST data (sample size n=25, mean SNR = 8.68), average precision = 95.8%, recall = 87.3%, and f1-score=90.0%; for HD-fMOST data (sample size n=25, mean SNR = 11.4), average precision = 98.1%, recall = 91.0%, and f1-score=93.3% (*Figure 4A*).

Notably, in low-SNR LSM data, background regions contain more artifactual signals (first panel in *Figure 4C*) due to similar intensity distributions between background and foreground points. In contrast, high-SNR datasets (fMOST and HD-fMOST) exhibit cleaner background features with distinct intensity separation between background noise and neurite signals (second and third panel in *Figure 4C*). This observation highlights the critical impact of SNR on reconstruction quality while simultaneously validating the robustness of PointTree, which is aided by the segmentation network, across diverse SNR conditions.

## Restrain error accumulation in the reconstruction

In order to achieve accurate axon reconstruction, it is essential to effectively suppress the snowballing accumulation of reconstruction errors. The performance of the minimal information flow tree (MIFT) in retraining the reconstruction errors is evaluated in this study. *Figure 5A* presents six 512×512 × 512 image blocks and their reconstructions using PointTree in the first column. The reconstruction fusing procedure is then performed on these axonal reconstructions (*Figure 5A*). By employing MIFT to revise the reconstructions and remove false connections between axons, reasonable reconstructions are achieved. In contrast, when the same fusion procedure is conducted without MIFT to revise the reconstruction, almost all axons are incorrectly connected together (bottom-right panel in *Figure 5A*).

We furthermore measure the enhancement in the reconstruction accuracy achieved by MIFT (*Figure 5B*). For the initial reconstructions from six image blocks, the average of f1-score is about 0.86. By using MIFT, the average of f1-score is above 0.8 for the reconstructions from two image blocks which are generated with the first fusion. In the second fusion (top-right panel in *Figure 5A*), the f1-score still keeps 0.79. In contrast, without MIFT, the first fusion leads to a drop of about f1-score of 0.3. After the second fusion, the f1-score is less than 0.2. We also present some reconstruction examples after two fusions in *Figure 5C*, which are close to the ground truth. These results suggest that the MIFT model takes consideration of the proper structure of axons and thus can restrain the error communications in the reconstruction fusion process.

## Long-range axonal projections reconstruction

We applied PointTree for long-range axon reconstruction. The testing image block has the size of 11226×8791 × 1486 voxels and includes axons from eight neurons (*Figure 6A*). We also used GTree to manually reconstruct these neurons as the ground-truth reconstruction (*Figure 6B*). Except for the labeling of training data for segmentation network and of the axon starting points of a single neuron, the whole reconstruction process is totally automatic. The results show PointTree successfully recovered the axonal morphology of these eight neurons without manual interference (*Figure 6C* and *Videos 1 and 2*), and we compared these reconstructions with ground truth (*Figure 6—figure supplement 1*). The average precision is above 85% and the average recall and f1-score are above 80% (*Figure 6E*). In addition, we presented the axon reconstructions from two image blocks (*Figure 6C1 and C2*) which include a large number of densely distributed axons. This reconstruction performance suggests that the point assignment and the minimal information flow tree mode, as the two key strategies in PointTree, perform well in long-range axonal reconstruction.

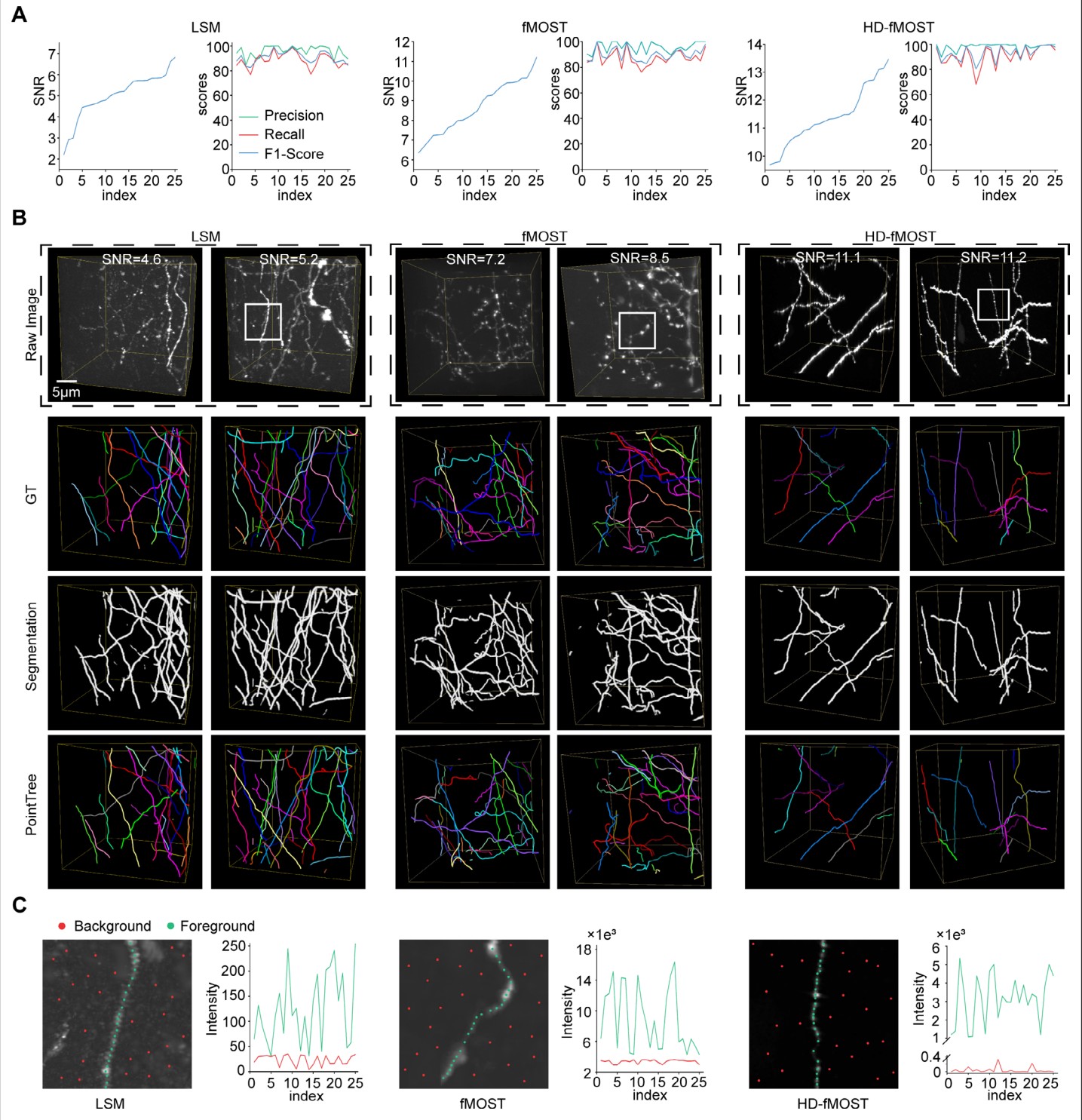

**Figure 4.** Reconstruction performance of PointTree across data with different signal-to-noise ratios. (**A**) Data blocks from light sheet microscopy (LSM), fluorescent micro-optical sectioning tomography (fMOST), and high-definition fluorescent micro-optical sectioning tomography (HD-fMOST) are selected. SNR and corresponding reconstruction scores with PointTree are drawn with line charts. Each dataset is of sample size n=25 and each data block size of 128×128 × 128. (**B**) shows reconstruction performance of PointTree on different datasets. (**C**) The zoomed-in view displays the region marked by white box in the first column of (**B**), with 25 foreground points and 25 background points sampled respectively. The signal intensities of both the foreground points and background points are plotted in the adjacent line charts.

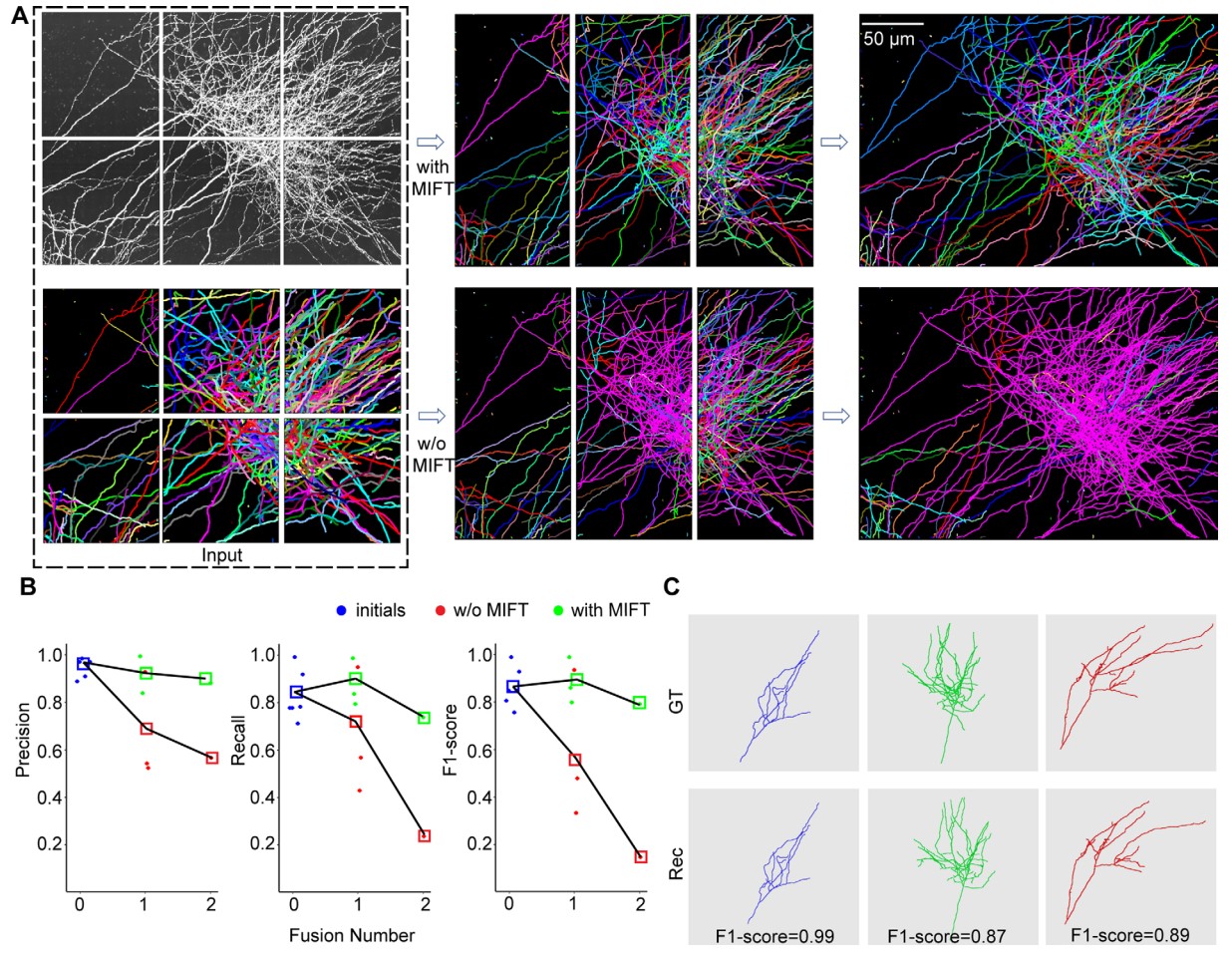

**Figure 5.** Minimal information flow tree effectively restrains the accumulation of reconstruction errors. (**A**) Reconstruction comparisons in the fusion process with MIFT and without MIFT are shown. Both image blocks and neurite reconstructions are displayed using maximum projection along the z-direction. Two fusion procedures are performed, and the final fusion reconstructions are presented in the third column. (**B**) The variation in reconstruction accuracy during the fusion process with MIFT and without MIFT is illustrated. Blue points represent the initial reconstruction accuracy from six image blocks, while green points and red points denote the merged reconstruction accuracy with MIFT and without MIFT, respectively. The squares represent the mean values of the evaluation indexes. (**C**) The skeletons of three neurite branches from the final merged reconstructions with MIFT are shown. Additionally, corresponding ground-truth reconstructions and reconstruction evaluations are also presented.

We also applied PointTree to process another 10739×11226 × 3921 image blocks collected with HD-fMOST system (*Zhong et al., 2021*). The high signal-to-noise ratio in this optical system results in a significantly extended dynamic range of the signal. PointTree can effectively deal with this case, and all 14 long-range projections are successfully reconstructed (*Figure 6—figure supplement 2*). The quantitative results suggest that the average f1-score is above 90% (*Table 1*).

Despite the need to solve multiple large-scale optimization problems, the reconstruction speed using PointTree is generally faster than the imaging speed. For instance, in a typical scenario involving 254 image blocks with 512×512 × 512 voxels, the total time required for reconstruction is approximately 44 min. Even for a larger dataset comprising 821 image blocks with 512×512 × 512 voxels and including a significant number of sparsely distributed neurites, the total time cost amounts to about 60 min (*Table 2*). It should be noted that the time cost does not increase linearly as data volume increases due to the influence of neurite density on overall reconstruction time. In summary, PointTree demonstrates remarkable speed in reconstructing long-range axons (*Video 3*).

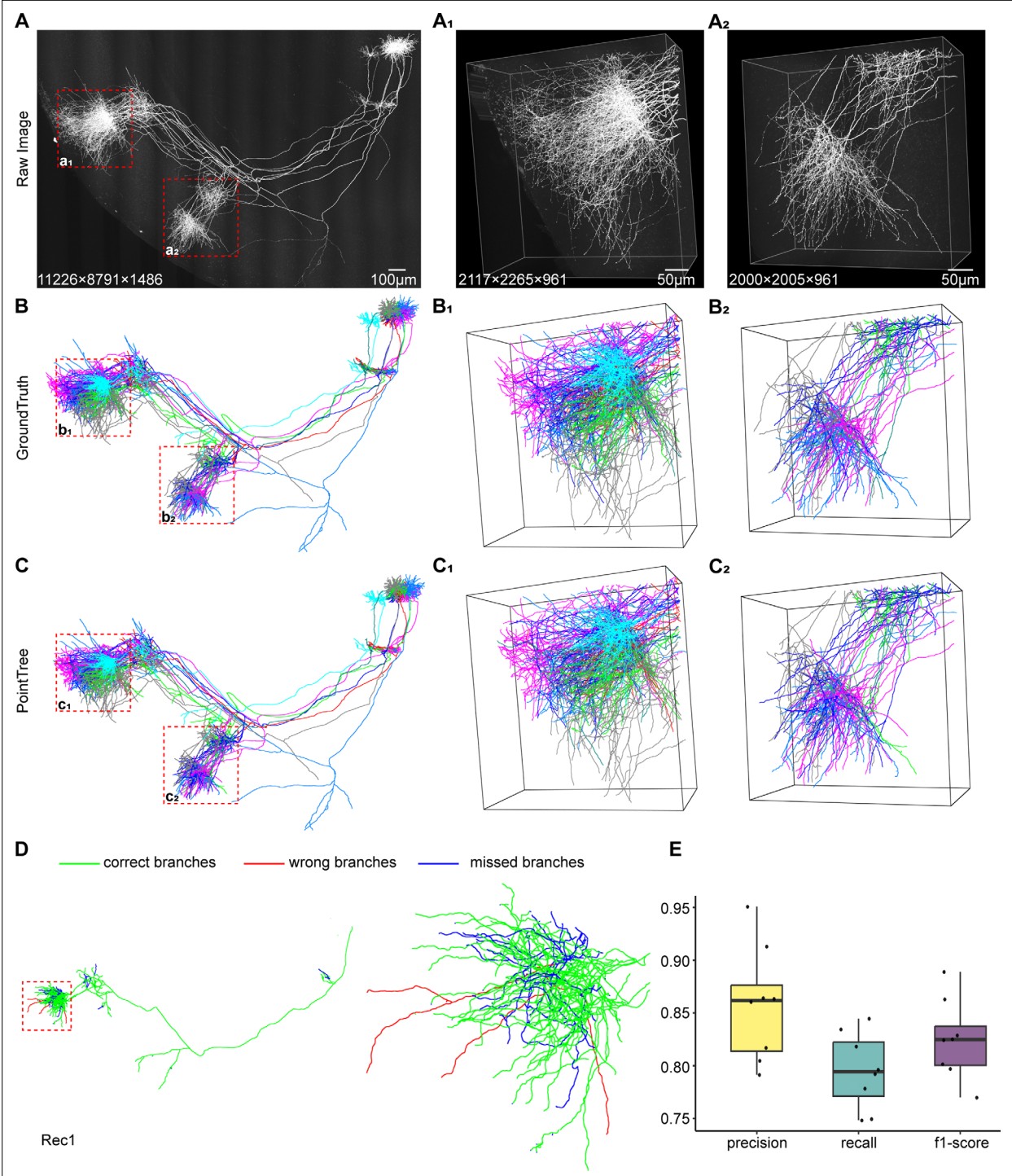

**Figure 6.** Long-range axonal reconstruction using PointTree. (**A**) The image block contains eight neurons in the ventral posteromedial thalamic region. The projection of these neurons includes a large number of densely distributed axons, which are enlarged in A$_1$ and A$_2$. (**B**) The reconstruction of the eight neurons is achieved by annotators with semi-automatic software GTree, serving as ground-truth reconstruction to evaluate automatic algorithms. The reconstructions B$_1$ and B$_2$ correspond to the image blocks A$_1$ and A$_2$. (**C**) Automatic reconstruction with PointTree results in reconstructions of the densely distributed axons, which are enlarged in C$_1$ and C$_2$. (**D**) A comparison between automatic reconstruction and ground-truth reconstruction of axonal projection for one neuron is shown. Green indicates consistent reconstruction, blue indicates missed branches, and red denotes branches from other neurons. (**E**) Quantitative analysis of long-range projections for these neurons is presented. Statistical information is displayed in boxes (n=8), the horizontal line in the box, the lower and upper borders of the box represent the median value, the first quartile (Q1) and the third quartile (Q3) respectively, the vertical black lines indicate 1.5×IQR, while black points represent the accuracy of the reconstructions for these neurons.

*Figure 6 continued on next page*

*Figure 6 continued*

The online version of this article includes the following figure supplement(s) for figure 6:

**Figure supplement 1.** Reconstruction of long-range axonal projections.

**Figure supplement 2.** Reconstruction from different datasets.

## Discussion

We have presented an automated method for reconstructing the long-range projections of neurons. In this study, we address the problem of mutual interference among densely distributed neurites and the cumulative error during reconstruction by designing a reconstruction method based on point set assignment and the minimal information flow tree, respectively. As a result, our approach enables accurate reconstruction of long-range neuron projections from hundreds of gigabytes of data. This advance significantly enhances the efficiency of whole-brain-scale neuron reconstruction, bridging the substantial gap between factory-level generation of whole-brain-scale neuronal imaging data and tens of hours required to reconstruct one neuron.

Our approach is performed on image foregrounds where the segmented neurites have a fixed radius approximately equal to the total size of the three voxels. In this case, we can estimate the total number of foreground points (voxels) and set a suitable number of columnar regions for ensuring the anisotropy of each columnar region, which is based on the fact that the union of columnar regions equals the foreground region. The anisotropy of the columnar regions will reduce the difficulty in establishing their connection. The requirement that all segmented neurites have a relatively fixed radius can be fulfilled. For all neurites, the value of their voxels decreases as these voxels deviate from the nearest centerline. The deep learning network is able to grasp this feature and segment only the neurite centerline and its neighborhood. Typically, in reconstructions of neurons whose projections are distributed over hundreds to thousands of GBs of data, less than GB-sized images with labels are needed as training data. The labeling process takes a few hours, which is negligible for semi-automatic reconstruction of all neurons in the whole volume images.

We propose a new reconstruction mode centered on point set assignment instead of the current reconstruction mode focused on skeleton extraction. In the current reconstruction paradigm, most deep networks are used to enhance the signal-to-noise ratio of neuronal images and do not address the issue of signal interference during skeleton extraction. In contrast, our reconstruction approach is based on directly processing the foreground points generated by the deep learning network. With continued advances in deep learning techniques, the generality and accuracy of image segmentation will be continuously enhanced, thereby significantly boosting the application scope of our method in various scenarios. Essentially, our method can be applied to any skeleton tracking-based application scenario and effectively eliminate dense signal interference.

Our method still generates a few reconstruction errors. This is due to the following three aspects. First, our method directly handles image foregrounds, which leads to reconstruction errors when some neurites with weak image intensities are not identified. Second, relying solely on foreground point information and rule-based judgment methods may generate some connection errors when establishing connections between neurites.

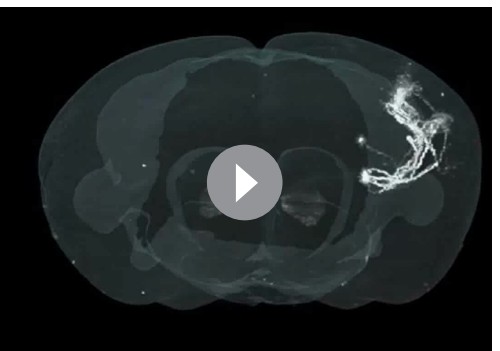

**Video 1.** Reconstructed long-range axonal projections and raw image data shown in *Figure 6*, individual axonal projections are delineated in different colors.
https://elifesciences.org/articles/102840/figures#video1

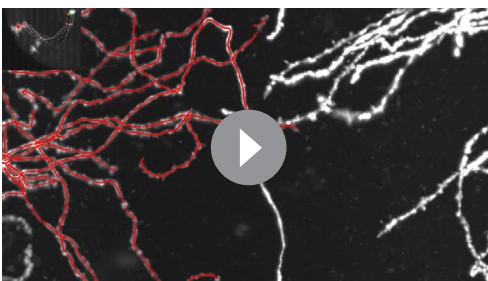

**Video 2.** Trace one of the reconstructed projections shown in *Figure 6*.
https://elifesciences.org/articles/102840/figures#video2

**Table 1.** Quantitative metrics comparing ground truth and reconstructed neurons are presented in *Figure 6—figure supplement 2*.

| ID | Precision | Recall | F1-Score |
|----|-----------|--------|----------|
| 1 | 1.00 | 0.92 | 0.95 |
| 2 | 1.00 | 1.00 | 1.00 |
| 3 | 0.98 | 0.76 | 0.86 |
| 4 | 1.00 | 0.82 | 0.90 |
| 5 | 1.00 | 0.77 | 0.87 |
| 6 | 1.00 | 0.92 | 0.96 |
| 7 | 0.96 | 0.75 | 0.84 |
| 8 | 1.00 | 0.87 | 0.93 |
| 9 | 1.00 | 0.82 | 0.90 |
| 10 | 1.00 | 0.96 | 0.98 |
| 11 | 1.00 | 0.99 | 0.99 |
| 12 | 1.00 | 0.77 | 0.87 |
| 13 | 1.00 | 0.90 | 0.95 |
| 14 | 0.99 | 0.87 | 0.93 |

Finally, the minimal information flow tree's fundamental assumption, that axons should be as smooth as possible, does not always hold true. In fact, real axons can take quite sharp turns (*Figure 1—figure supplement 3*) leading the algorithm to erroneously separate a single continuous axon into disjoint fibers (*Figure 1—figure supplement 3*). Therefore, for the automatic reconstruction of neurons on a brain-wide scale, further work is needed to enhance the imaging intensity and incorporate soma shapes and raw image signals for neurites connection recognition.

## Materials and methods
### Data collections
All animal experiments followed procedures approved by the Institutional Animal Ethics Committee of the Huazhong University of Science and Technology. The test datasets are collected through the preparation of two kinds of samples. For one C57BL/6 male mouse, 100 nl AAV-Cre virus and 100 nl of AAV-EF1α-DIO-EYFP virus were injected into the VPM nucleus at the same time. 21 days later, the chemical sectioning fluorescence tomography (CSFT) system (*Wang et al., 2021*) was used to acquire imaging data (*Figures 1–6*), more details can be seen in the reference (*Zhang et al., 2021*). For one C57BL/6 J male mouse, 100 nl of AAV-YFP was injected into the motor area. 21 days later, high-definition fluorescent micro-optical sectioning tomography (HD-fMOST) was used to acquire imaging data (*Zhong et al., 2021*; *Figure 6—figure supplement 2*).

### Generation of foreground points
Our reconstruction method performs on the image foregrounds. Here, we used UNet3D (*Çiçek et al., 2016*) for image stacks segmentation without network structure modification. The detailed information about UNet3D can be found in the reference (*Çiçek et al., 2016*). Considering the requirement

**Table 2.** Time cost of three modules in the entire reconstruction for two testing datasets shown in *Figure 6*, *Figure 6—figure supplement 2*.

| block number (size: 512×512 × 512) | Points clustering (mins) | Clusters connection (mins) | Reconstruction merging (mins) |
|---|---|---|---|
| 254 | 23 | 18 | 3 |
| 821 | 22 | 35 | 3 |

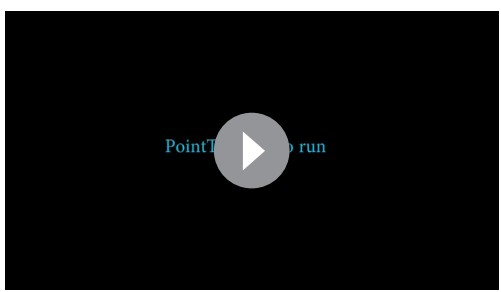

**Video 3.** Example run of PointTree on Windows.
https://elifesciences.org/articles/102840/figures#video3

that the network output, the segmented neurites, have the relatively fixed radius, we calculate the distance field of the neurite's skeleton as the ground truth for supervising the network. Initially, the semi-automatic software GTree was utilized to extract the neurite skeleton and subsequently interpolate the skeleton points. The interpolation operation ensured that the distance between any skeleton point and its nearest point was less than 1 μm. Subsequently, the interpolated skeleton points were used as centers to mark spherical regions with a radius of 5 voxels. These spherical regions served as candidate areas for foreground. Within these candidate areas, the distance from each point to its nearest interpolated skeleton point was calculated. Finally, the distances are mapped into Gaussian kernel distances, which form the Gaussian density map. This map normalized by maximum value leads to the distance field map to supervise UNet3D output.

In the training stage, Adam optimizer is used with an initial learning rate at 3e-4. The input image size is 128×128 × 128. Batch size is set to 1, the L1-norm is used as loss function to train the network. We presented the reconstructions from two kinds of fMOST datasets. One is from the reference (*Zhang et al., 2021*) and the other is from the reference (*Zhong et al., 2021*). Therefore, we created two sets of training data, each consisting of 20 512×512 × 512 image blocks (each divided into 64 image blocks of size 128×128 × 128). In each set, 10 image blocks contain densely distributed neurites, while the other 10 blocks contain sparsely distributed neurites. In the predicting stage, we applied the threshold operation to the distance field image. The voxels whose values are more than 0.5 are regarded as the foreground points.

## Neuron Reconstruction based on Points assignment

For the image stack, we allocated the foreground points to their respective neurites and established connections between neurites by constructing three optimization models: (1) the constrained Gaussian mixture model divides the foreground points into a set of points, each of which has a column shape; (2) the minimum-volume covering ellipsoids model extracts the features of the column-shaped point set; (3) the 0–1 assignment optimization model establishes connections between the column-shaped point sets, resulting in the shapes of individual neurites, and then builds connections between the reconstructed neurites.

## Constrained Gaussian mixture model

The three-dimensional Gaussian function exhibits an ellipsoidal shape in space, which we have utilized to approximate the columnar shape of local neurites. In this study, Gaussian distribution mixture functions with $K$ components are employed to approximate the shape of all neurites in an image block. The component number $K$ is obtained by point density and will be discussed later. Given the foreground points $x_1, x_2, \cdots, x_n$, for each foreground points $x_i$, the probability density function $P(x_i)$ is calculated as follows:

$$P(x_i) = \sum_{j=1}^{K} \pi_j N(x_i|\mu_j, \Sigma_j) \tag{1}$$

Here, $N(x_i|\mu_j, \Sigma_j)$ is the Gaussian density function with mean value $\mu_j$ and covariance matrix $\Sigma_j$. Weight $\pi_j$ is the regularization parameter. $N(x_i|\mu_j, \Sigma_j)$ is given by the formula:

$$N(x_i|\mu_j, \Sigma_j) = \frac{1}{2\pi^{3/2}|\Sigma_j|^{1/2}} e^{-\frac{1}{2}(x_i-\mu_j)^T \Sigma_j^{-1}(x_i-\mu_j)} \tag{2}$$

Based on probability density function, the conditional probability can be computed as:

$$p_{i,j} = P\left(x_i|cluster_j\right) = \frac{\pi_j N\left(x_i|\mu_j, \Sigma_j\right)}{\sum_{j=1}^{K} \pi_j N\left(x_i|\mu_j, \Sigma_j\right)} \qquad j = \left(1, 2, ..., K\right) \tag{3}$$

Here, $p_{i,j}$ is the conditional probability for $x_i$ to assign to the $j$-th cluster. If $p_{i,k}$ is the maximum value among $\{p_{i,1}, ... p_{i,K}\}$, the foreground point $x_i$ will be assigned to the $k$-th cluster. All the points assigned to the $k$-th cluster form a columnar region. Considering that both the number of foreground points and component number are large, we have added some constrained conditions for Gaussian mixture model as follows:

$$\sum_{j=1}^{K} \pi_j = 1 \tag{4}$$

$$I\left(\mu_j\right) \geq \varepsilon_0, |\Sigma_j| \leq \varepsilon_1 \tag{5}$$

$\sum_{j=1}^{K} \pi_j = 1$ refers to the fact that the total probability distribution normalizes to 1. $I\left(\cdot\right)$ represents the signal intensity from segment image, $\varepsilon_0$ is the minimum signal intensity of foreground points and is set to 128 in the algorithm. $I\left(\mu_i\right) \geq \varepsilon_0$ restrain the center of the Gaussian distribution to be a foreground point. $|\Sigma_j| \leq \varepsilon_1$ restrain the determinant of the covariance matrix which controls the suitable number of foreground points for each columnar region. $\varepsilon_1$ is set to the cube of three times the average diameter of neurite.

Maximum likelihood is employed to estimate the parameters of Gaussian mixture model and the final optimization problem is formed as follows:

$$\left(\pi_j^*, \mu_j^*, \Sigma_j^*\right)_{j=1,2,\cdots,K} = \arg\max \prod_{i=1}^{n} P\left(x_i\right) = \arg\max \prod_{i=1}^{n} \left(\sum_{j=1}^{K} \pi_j N\left(x_i|\mu_j, \Sigma_j\right)\right) \tag{6}$$

$$s.t. \sum_{j=1}^{K} \pi_j = 1, I\left(\mu_j\right) \geq \varepsilon_0, |\Sigma_j| \leq \varepsilon_1 \tag{7}$$

In solving this optimization problem, we employ peak density algorithm (*Wei et al., 2023*) to compute density for each foreground points and sort them in descending order. We first select a point as a seed point, and the foreground points within a radius of 5 centered on it will be excluded. Then we continue selecting seed points until all foreground points are either selected or excluded. The selected $K$ seed points represent the initial $K$ components. We select signal points from the median (based on density) to both sides as seed points, which can decrease the situations that seed points lie in the center of a crossover or the edge of neurites. This strategy can make the generated columnar regions be more reasonable. The positions of the $K$ seed points are set to the initial $\left(\mu_1, \mu_2, \cdots, \mu_K\right)$. The initial setting of the covariance matrix is the identity matrix. The constrained Gaussian mixture model was solved by the EM algorithm (*McLachlan and Krishnan, 2007*), the EM algorithm is divided into two steps:

E-step: For each point $x_i$, compute its probability within each Gaussian distribution using the probability density function:

$$p_{i,j} = \frac{\pi_j N\left(x_i|\mu_j, \Sigma_j\right)}{\sum_{j=1}^{K} \pi_j N\left(x_i|\mu_j, \Sigma_j\right)} \tag{8}$$

M-step: Update the mean value, covariance matrices, and weight vectors.

$$\pi_j = \frac{\sum_{i=1}^{n} p_{i,j}}{n} \tag{9}$$

$$\mu_j = \frac{\sum_{i=1}^{n} p_{i,j} x_i}{\sum_{i=1}^{n} p_{i,j}} \tag{10}$$

$$\Sigma_j = \frac{\sum_{i=1}^{n} p_{i,j} \left(x_i - \mu_j\right) \left(x_i - \mu_j\right)^T}{\sum_{i=1}^{n} p_{i,j}} \tag{11}$$

Besides, the constrained Gaussian mixture model possesses additional constraints: $I\left(\mu_j\right) \geq \varepsilon_0$ and $|\Sigma_j| \leq \varepsilon_1$. After finishing the M-step, $\mu_j$ with $I\left(\mu_j\right) < \varepsilon_0$ are selected. Eigenvalue decomposition is applied on $\Sigma_j$ and obtains eigenvalues $(\gamma_1, \gamma_2, \gamma_3)$ in descending order and eigenvectors $(v_1, v_2, v_3)$. $\mu_j$ is updated along $v_1$ and $-v_1$ to generate two new clusters with mean value and covariance matrices $\left(u_{j,1}, \Sigma_{j,1}\right)$ and $\left(u_{j,2}, \Sigma_{j,2}\right)$ as follows:

$$u_{j,1} = u_j + v_1 \cdot \frac{\gamma}{2} \tag{12}$$

$$u_{j,2} = u_j - v_1 \cdot \frac{\gamma}{2} \tag{13}$$

$$\Sigma_{j,1} = \frac{\sum_{i=1}^{n} p_{i,j}\left(x_i - \mu_{j,1}\right)\left(x_i - \mu_{j,1}\right)^T}{\sum_{i=1}^{n} p_{i,j}} \tag{14}$$

$$\Sigma_{j,2} = \frac{\sum_{i=1}^{n} p_{i,j}\left(x_i - \mu_{j,2}\right)\left(x_i - \mu_{j,2}\right)^T}{\sum_{i=1}^{n} p_{i,j}} \tag{15}$$

For $\Sigma_j > \varepsilon_1$, it will be updated as follows:

$$\Sigma_j' = \frac{\varepsilon_1}{\Sigma_j}\Sigma_j \tag{16}$$

Iteration of E-step and M-step will continue until the $k$-th result $\left\{\mu^k, \Sigma^k\right\}$ and $(k$-1$)$-th result satisfy the stopping criteria:

$$\left\|\frac{u^k - u^{k-1}}{u^{k-1}}\right\| < \varepsilon \text{ and } \left\|\frac{\Sigma^k - \Sigma^{k-1}}{\Sigma^{k-1}}\right\| < \varepsilon \tag{17}$$

Here the division represents element-wise division and $\|\cdot\|$ denotes $L_2$-norm and $\varepsilon$ is set to 0.01.

## Shape characterization of columnar regions

After deriving the columnar regions through solving the constrained Gaussian mixture model, it is imperative to characterize their geometric shape (terminals and centerlines). For this purpose, we calculate the minimum-volume ellipsoids that can fully encompass each individual columnar region. For $c \in R^3$, $Q \in S_{++}^3$, a three-dimensional ellipsoid can be defined as follows *Sun and Freund, 2004*:

$$E_{c,Q} := \left\{x \in R^3 \mid \left(x - c\right)^T Q\left(x - c\right) \leq 1\right\} \tag{18}$$

Here, $c$ is the center of ellipsoid, $Q$ represents the geometric shape, $S_{++}^3$ denotes the convex cone of 3×3 symmetric positive definite matrices. The volume of $E_{c,Q}$ is given by the formula:

$$Volume\left(E_{c,Q}\right) = \frac{\pi^{3/2}}{\Gamma\left(3/2 + 1\right)}\frac{1}{\sqrt{det\left(Q\right)}} \tag{19}$$

Here, $\Gamma\left(\cdot\right)$ is the standard gamma function of calculus, $det\left(Q\right)$ means the determinant of matrix $Q$. Minimizing the volume of $E_{c,Q}$ is equivalent to minimizing $det\left(Q^{-1/2}\right)$. Therefore, for a columnar region with foreground points $P\left\{x_1, x_2, \ldots x_m\right\}$, we define the target function as follows:

$$P1: \left(c^*, Q^*\right) = \arg\min_{c,Q} det\left(Q^{-1/2}\right) \tag{20}$$

$$s.t. \left(x_i - c\right)^T Q\left(x_i - c\right) \leq 1, i = 1, 2 \ldots m \tag{21}$$

$$c \in CHull\left(P\right), Q \in S_{++}^3 \tag{22}$$

Here $c \in CHull\left(P_i\right)$ restrain the solved center of ellipsoid to locate within the smallest convex hull formed by the clustering points. To solve this problem, a variable substitution $A = Q^{1/2}$ and $y = Q^{1/2}c$ were applied to *Equation 20* and *Equation 21*, the original problem *P*1 can be transformed into a convex optimization problem as follows:

$$P2: \left(A^*, y^*\right) = \underset{A,y}{\arg\min} - \ln\det\left(A\right) \tag{23}$$

$$s.t. \left(Ax_i - y\right)^T \left(Ax_i - y\right) \leq 1, \quad i = 1, 2, ..., m \tag{24}$$

$$A \in S_{++}^3 \tag{25}$$

Through adding the logarithmic barrier function, we can obtain the following formula:

$$P3: \left(A^*, y^*, \theta^*\right) = \underset{A,y,\theta}{\arg\min} - \ln\det\left(A\right) - \theta \sum_{i=1}^m \ln\left(z_i\right) \tag{26}$$

$$s.t. \left(Ax_i - y\right)^T \left(Ax_i - y\right) + z_i = 1, \quad i = 1, 2, ..., m \tag{27}$$

$$A \in S_{++}^3, z_i > 0 \tag{28}$$

As $\theta$ varies in the interval $(0, \infty)$, the solution of $P3$ changes. When $\theta$ approaches 0, the optimal solution of $P3$ tends to the optimal solution of $P2$. By adding the dual multipliers $d_i$ which satisfies $d_i \cdot z_i = \theta$, the optimality conditions can be written as:

$$\sum_{i=1}^m d_i \left[\left(Ax_i - y\right) x_i^T + x_i \left(Ax_i - y\right)^T\right] - A^{-1} = 0 \tag{29}$$

$$\sum_{i=1}^m d_i \left(y - Ax_i\right) = 0 \tag{30}$$

$$\left(Ax_i - y\right)^T \left(Ax_i - y\right) + z_i = 1 \, i = 1, 2, ..., m \tag{31}$$

$$\sum_{i=1}^m d_i \cdot z_i = \theta, \quad i = 1, 2, ..., m \tag{32}$$

$$d_i, z_i \geq 0 \tag{33}$$

At this point, the error between the solution of the system of equations and the optimal solution of $P3$ is less than $d^T z$. Through **Equation 30**, the explicit expression for solving $y$ can be obtained as follows:

$$y = \frac{AXd}{e^T d} \tag{34}$$

Here, $X$ stands for a $3 \times m$ matrix $[x_1|x_2|...|x_m]$, $e$ stands for vector of ones $(1, 1, ..., 1)_{1 \times m}^T$ and $d$ stands for $(d_1, d_2, ..., d_m)_{1 \times m}^T$. Substitute **Equation 34** into **Equation 29**, the equation for matrix $A$ can be obtained by:

$$\left(XDX^T - \frac{Xdd^T X^T}{e^T d}\right) A + A \left(XDX^T - \frac{Xdd^T X^T}{e^T d}\right) = A^{-1} \tag{35}$$

Here, $D$ stands for a $m \times m$ diagonal matrix $Diag\left(d_1, d_2, ..., d_m\right)$. And the explicit expression for $A$ is formed as

$$A = A\left(d\right) = \left[2\left(XDX^T - \frac{Xdd^T X^T}{e^T d}\right)\right]^{-1/2} \tag{36}$$

And explicit expression for $y$:

$$y = \frac{\left[2\left(XDX^T - \frac{Xdd^T X^T}{e^T d}\right)\right]^{-1/2} Xd}{e^T d} \tag{37}$$

Through substituting the above two equations to the system of **Equations 29-33**, variables $A$ and $y$ are eliminated. The following system of equations with only variables $d$ and $z$ can be obtained:

$$f\left(d\right) + z - e = 0 \tag{38}$$

$$Dz - \theta e = 0 \tag{39}$$

$$d_i, z_i \geq 0 \tag{40}$$

Here, $f(d)$ is nonlinear function of variable $d$:

$$f_i(d) = \left(x_i - \frac{Xd}{e^T d}\right)\left[2\left(XDX^T - \frac{Xdd^T X^T}{e^T d}\right)\right]^{-1} \cdot \left(x_i - \frac{Xd}{e^T d}\right) \quad i = 1, 2, ..., m \tag{41}$$

For a fixed barrier parameter $\theta$, we employ Newton's method to solve the system of equations. We use $\nabla_d f(d)$ to represent the Jacobian matrix of $f(d)$. Thus, the Jacobian matrix of the system of equations can be computed as follows:

$$\begin{bmatrix} \nabla_d f(d) & I \\ Z & D \end{bmatrix} \tag{42}$$

And the Newton's direction is written as:

$$\Delta(d) = \left(\nabla_d f(d) - D^{-1} Z\right)^{-1} \left(h_1 - D^{-1} h_2\right) \tag{43}$$

$$\Delta(z) = D^{-1} h_2 - D^{-1} Z \left(\nabla_d f(d) - D^{-1} Z\right)^{-1} \left(h_1 - D^{-1} h_2\right) \tag{44}$$

$$h_1 = e - z - f(d), \quad h_2 = \theta e - Dz, \tag{45}$$

With initial $(d_0, z_0)$, iterate with $(d_n, z_n) = (d_{n-1}, z_{n-1}) + \tilde{\beta}\left(\Delta(d_{n-1}), \Delta(z_{n-1})\right)$ to obtain the final optimal solution, $\tilde{\beta}$ represents the Newton's step. Detailed process can see the pseudo code as follows:

---

Algorithm 1. **Compute Newton's direction.**

---

**Input:** $(d, z, \theta)$ satisfying $d, z > 0$, $\theta \geq 0$

1. $\quad A^{-2}(d) = \left[2\left(XDX^T - \frac{Xdd^T X^T}{e^T d}\right)\right]$

2. $\quad \Sigma(d) = \left(X - \frac{Xde^T}{e^T d}\right) A^2(d) \left(X - \frac{Xde^T}{e^T d}\right)$

3. $\quad \nabla_d f(d) = -2\left(\frac{\Sigma(d)}{e^T d} + \Sigma(d) \circ \Sigma(d)\right)$

4. $\quad (\Delta(d), \Delta(z)) = \left(\left(\nabla_d f(d) - D^{-1} Z\right)^{-1}\left(h_1 - D^{-1} h_2\right), D^{-1} h_2 - D^{-1} Z\Delta(d)\right)$

---

**Output:** $(\Delta(d), \Delta(z))$

---

Algorithm 2. **Process of solving P2.**

---

**Input:** $\{x_1, x_2, ..., x_m\}$

1. $r = 0.99$, $(d_0, z_0) = \left(\frac{3}{2m}e, e - f(d_0)\right)$

2. $E = -\det(A(d))$

3. while $\left(|e - f(d) - z| > \varepsilon_1 \text{ or } \frac{d^T z}{E} > \varepsilon_2\right)$

4. $\qquad \theta = \frac{d^T z}{10m}$

5. $\qquad (\Delta(d), \Delta(z)) = Compute\_Newton\_direction(d, z)$

6. $\qquad \bar{\beta} = \max\{\beta \mid (d, z) + \beta(\Delta(d), \Delta(z) \geq 0)\}$

7. $\qquad \tilde{\beta} = \min(\bar{r}\bar{\beta}, 1)$

8. $\qquad (d, z) = (d, z) + \tilde{\beta}(\Delta(d), \Delta(z))$

9. $\qquad E = -\det(A(d))$

---

**Output:** $Q = A(d)^2, c = A(d)^{-1} y(d)$

---

With the solved optimal solution of $(Q, c)$, we then check whether $c$ is located within the convex hull of the input point set $\{x_1, x_2, ..., x_m\}$. If it is not, a constrained Gaussian mixture model will be applied to partition it into two subsets and solve the minimum-volume covering ellipsoids problem again in the two subsets. Through solving the above minimum-volume covering ellipsoids problem, we can characterize the columnar regions more accurately.

Note that from constrained GMM, each cluster has the corresponding mean and covariance matrix of points in the cluster. These two values essentially describe the shape of the cluster. However, if these two values directly replace $c^*$ and $Q^*$, the exported ellipsoid may only encompass a part of points in the cluster. For covering all points in the cluster, all elements in the covariance matrix are needed to be proportionally enlarged, but the volume of the corresponding ellipsoid is not minimum. These two cases will reduce the accuracy of the connections between clusters, that is columnar regions. So, we introduce the minimum-volume covering ellipsoid model to extract the shape of columnar region.

## Skeleton generation using 0-1 assignment model

The 0–1 assignment model (***Volgenant, 1996***) can robustly and accurately establish connections between particles in live-cell imaging (***Jaqaman et al., 2008***). It is particularly effective in handling cases where particles are densely distributed, merged, or split. We analogize column regions to particles and apply the 0–1 assignment model to build the connections between column regions. For the $i$-th columnar region, the center and the two endpoints of the longest axis of its minimum-volume covering ellipsoid are denoted by $c_i, t_{i,0}, t_{i,1}$. The direction refers to the pointing of the center point towards $t_{i,k}$, $k$ equal to 0 or 1. According to the direction and the endpoints, we design the cost matrix for building the 0–1 assignment model.

$$
C = \begin{bmatrix}
c\left(t_{1,0},t_{1,0}\right) & c\left(t_{1,0},t_{1,1}\right) & \cdots & c\left(t_{1,0},t_{n,1}\right) & & \\
c\left(t_{1,1},t_{1,0}\right) & c\left(t_{1,1},t_{1,1}\right) & \cdots & c\left(t_{1,1},t_{n,1}\right) & D & \\
\vdots & \vdots & \vdots & \vdots & & \\
c\left(t_{n,1},t_{1,0}\right) & c\left(t_{n,1},t_{1,1}\right) & \cdots & c\left(t_{n,1},t_{n,1}\right) & & \\
& & & & & \\
& & D & & D &
\end{bmatrix}_{4n \times 4n}
\tag{46}
$$

$$
c\left(t_{i,i0},t_{j,j0}\right) = \begin{cases}
100 & if\ (i = j) \\
\dfrac{norm\left(t_{i,i0},t_{j,j0}\right)}{\left(0.5 \times \left(\dfrac{\theta\left(t_{i,i0},t_{j,j0}\right)}{3} + 1.001\right)\right)^4} & if\ (i \neq j)
\end{cases}
\tag{47}
$$

$$
\theta\left(t_{i,i0},t_{j,j0}\right) = \left\langle dir\left(c_i,t_{i,i0}\right), dir\left(c_i,t_{j,j0}\right)\right\rangle + \left\langle dir\left(c_j,t_{j,j0}\right), dir\left(c_j,t_{i,i0}\right)\right\rangle \\
- \left\langle dir\left(c_i,t_{i,i0}\right), dir\left(c_j,t_{j,j0}\right)\right\rangle
\tag{48}
$$

Here, $D$ is $2n \times 2n$ auxiliary matrix all elements of which are all set 100. Both $i0$ and $j0$ in **Equation 47** are equal to 0 or 1, labeling the two endpoints of the longest axis of the ellipsoid. $norm\left(t_{i,i0},t_{j,j0}\right)$ represents the Euclidean distance between $t_{i,i0}$ and $t_{j,j0}$. $\theta\left(t_{i,i0},t_{j,j0}\right)$ describes the angle between two ellipsoids, that is two columnar regions. $dir\left(c_i,t_{i,i0}\right)$ represents the line from point $c_i$ to $t_{i,i0}$. $\left\langle dir\left(c_i,t_{i,i0}\right), dir\left(c_i,t_{j,j0}\right)\right\rangle$ represents cosine angle between the two lines. The threshold of 100 in D in **Equation 46** and **Equation 47** is an experimental value designed to ensure that the terminal points of neurites do not connect to more than one other terminal point.

After setting the cost matrix, the 0–1 assignment problem is defined as follows:

$$
A = \arg\min_A \sum_{i=1}^{4n}\sum_{j=1}^{4n} A_{ij}C_{ij}
\tag{49}
$$

$$
s.t. \sum_{i=1}^{4n} A_{i,j} = 1\ \left(j = 1,2,\cdots,4n\right)
\tag{50}
$$

$$
\sum_{j=1}^{4n} A_{i,j} = 1\ \left(i = 1,2,\cdots,4n\right)
\tag{51}
$$

Here, $A$ represents the connectivity matrix between different terminals of columnar regions: if $A_{i,j} = 1$, then establish connection between terminal $i$ and terminal $j$, if $A_{i,j} = 0$, then establish no connection between terminal $i$ and terminal $j$. $\sum_{i=1}^{4n} A_{i,j} = 1\ \left(j = 1,2,\cdots,4n\right)$ and $\sum_{j=1}^{4n} A_{i,j} = 1\ \left(i = 1,2,\cdots,4n\right)$ restrain each terminal from establishing connection with at most one other terminal. The Lapjv algorithm (**Volgenant, 1996**) is utilized to solve this optimization problem and the shapes of individual neurites in block images are formed. Furthermore, we employ the region growing method to generate skeletons from the reconstructed shape, achieving the neurites reconstruction from individual image blocks.

## Minimal information flow tree for revising the reconstruction

The minimal information flow tree model is designed to modify the topology of skeletons, eliminate incorrect connections, and decompose them into multiple branches. When given an input skeleton file such as the swc file (**Cannon et al., 1998**), we convert it into a binary tree structure with the following steps.

### Step 1

select the neurite skeleton $S_1$. $S_1$ has the largest length in the neurite skeletons that connect with each other. One of its terminal nodes is recorded as the head node $n_1$.

## Step 2

generate the initial tree structure. Starting at head node $n_1$, search the linking nodes along the skeleton $S_1$, denoted by $n_1^{s_1}, n_2^{s_1}, \cdots, n_{k_1}^{s_1}$. The topology structure is $n_i \rightarrow leftnode = n_{i+1}^{s_1}$.

## Step 3

generate new structure induced by the linking node $n_1^{s_1}$. $n_1^{s_1}$ is regarded as the head node and its corresponding neurite skeleton is denoted by $S_2$. Let $n_1^{s_2}, n_2^{s_2}, \cdots, n_{k_2}^{s_2}$ represent the linking nodes in skeleton $S_2$. The corresponding topology structure is $n_1^{s_1} \rightarrow rightnode = n_1^{s_2}$, $n_i^{s_2} \rightarrow leftnode = n_{i+1}^{s_2}$.

## Step 4

repeat the operation in **Step 3** for dealing with the linking nodes $n_2^{s_1}, \cdots, n_{k_1}^{s_1}$. The corresponding topology structures are added into the total tree structure. After obtaining the tree structures induced by linking nodes in $S_1$, use the operation in **Step 3** to generate the tree structures induced by linking nodes in $S_2$. Continue in this manner until all linking nodes have been processed.

To gain a better understanding of the above process, we have provided a demonstration of how to generate the corresponding binary tree from the skeletons of neurites (***Figure 1—figure supplement 4***).

For the skeletons of neurites in an image block, the corresponding number of binary tree structures will be generated. We use the MIFT model to merge or split these binary structures. Suppose that an image stack contains $m$ skeletons all of which have $K$ nodes, denoted by $n_1, \cdots, n_{K-1}, n_K$. The connections among these nodes are stored in a matrix $W$ with $K \times K$ elements. $W_{i,j} = 0$ indicates that there is no connection between node $i$ and node $j$. $W_{i,j} = -1$ indicates that $j \rightarrow headnode = i$, $W_{i,j} = -2$ indicates that $j \rightarrow leftnode = i$, $W_{i,j} = -3$ indicates that $j \rightarrow rightnode = i$.

The information flow can be computed as follows:

$$W^* = \arg\min_W \sum_{i=1}^{K} f(W, n_i) \tag{52}$$

$$f(W, n_i) = \cos\left(\theta\left(n_i \rightarrow headnode, n_i, n_i \rightarrow leftnode\right)\right) \tag{53}$$

Here, the optimization objective function in ***Equation 53*** is called information flow. $\theta(\cdot)$ is the angle between flow from $n_i \rightarrow headnode$ to $n_i$ and flow from $n_i$ to $n_i \rightarrow leftnode$. To minimize the optimization problem while ensuring that the topology matrix $W$ does not exhibit abnormal values, we adopt the strategy of dynamic programming to update the topology matrix $W$. Briefly, we calculate the other two possible angles $\theta(n_i \rightarrow headnode, n_i, n_i \rightarrow rightnode)$ and $\theta(n_i \rightarrow leftnode, n_i, n_i \rightarrow rightnode)$ at the first linking node $n_i$. The minimum information flow is selected, and $W$ is updated. Following the updated $W$, the next branching node is found and information flow and $W$ is updated. The updating process iterates until all nodes are updated. The final root nodes $\{r_1, r_2, ..., r_m\}$ are obtained (node satisfies $W(r_t, i) = 0$ or $-1$ $(i = 1, ...n)$ is set root node). The pseudo-code for solving the optimization problem is provided below:

Algorithm 3. **Generation of Minimal Information Flow Tree.**

```
# Graph defines tree topology of the nodes, t_node->left represents the left child node of t_node, t_node->right
represents the right child node of t_node, t_node->head represents the head node of t_node.
```
Input: $N$: $\{N_0, N_1, ..., N_k\}$, Graph head: $\{N_0\}$
 $Set = \{N_0\}$
 While $|Set| > 0$:
 $t\_node = Set\,[0]$
 `# calculate three possible information flow`
 $res = calc\_three\_directions\,(t\_node)$
 if $(res == 0)$:
 `# maintain original structure.`
 $Set\,[0] = t\_node->left$
 $Set.push\_back\,(t\_node->right)$
 if $(res == 1)$:
 `# change the position of t_node's two child nodes.`
 $Exchange\_child\,(t\_node)$
 $Set\,[0] = t\_node->left$
 $Set.push\_back\,(t\_node->right)$
 if $(res == 2)$:
 `# Information flows from t_node->left to t_node->right, update the structure along t_node->left and`
 `t_node->head, generate new head if possible.`
 $New\_node = Reverse\_head\,(t\_node)$
 $Set\,[0] = New\_node$
Output: $N$: $\{N_0, N_1, ..., N_k\}$, Graph head: $\{N_0', N_1', ..., N_m'\}$.

Please note that the model has the capability to merge binary trees. When two branches of neurites have identifiable root nodes, and one root node is in close proximity to the skeleton points on the other branch of neurites, the root node does not contribute to the calculation of information flow without fusion. However, after fusion, the root node becomes a linking node in the other branch of neurites, resulting in an additional negative information flow value. In this merging process, a threshold is required to be set. When the minimum distance between the root node of a branch of neurites and the skeleton point of the other branch of neurites is less than 8 for individual image blocks or less than 8,12,16 for fused image blocks respectively, these two branches are merged. When splitting a branch of neurites, the minimal information flow tree model is also applied to both individual and fused image blocks.

## The fusion of neurites reconstruction

By using the MIFT model to revise the neurites reconstruction in individual image blocks, the root nodes and leaf nodes of a branch of neurites can be extracted directly. Here, we use a 0–1 assignment model to merge the reconstructions between two adjacent image blocks. For two adjacent image blocks $P$ and $Q$, the neurite skeleton nodes which locate near the common boundary are extracted as $\{p_1, p_2, ...p_m\}$, $\{q_1, q_2, ...q_n\}$ and the cost matrix is constructed as follows:

$$
C = \begin{bmatrix}
c\,(p_1, q_1) & \cdots & c\,(p_1, q_n) & \\
\vdots & \ddots & \vdots & D_{m \times m} \\
c\,(p_m, q_1) & \cdots & c\,(p_m, q_n) & \\
 & D_{n \times n} & & D_{n \times m}
\end{bmatrix}_{(m+n) \times (m+n)}
\tag{54}
$$

$$
c\,(p_i, q_j) = d\,(p_i, q_j) \times \left(2 - \theta\,\left(L\,(p_i), L\,(q_j)\right)\right)
\tag{55}
$$

Here, $D_{m \times m}$, $D_{n \times n}$, $D_{n \times m}$ are auxiliary matrix which the values are all set 20. $d\,(p_i, q_j)$ represents the Euclidean distance between terminal $p_i$ and $q_j$. $L\,(p_i)$ and $L\,(q_j)$ are fitted lines from the skeleton points near $p_i$ and $q_j$. $\theta\,\left(L\,(p_i), L\,(q_j)\right)$ represents the cosine value of their angle. Thus, the 0–1 assignment problem is formed as follows:

$$
A = \arg\min_A \sum_{i=1}^{m+n} \sum_{j=1}^{m+n} A_{i,j} \cdot C_{i,j}
\tag{56}
$$

$$s.t. \sum_{i=1}^{m+n} A_{i,j} = 1 \ (j = 1, 2, \ldots m + n) \tag{57}$$

$$\sum_{j=1}^{m+n} A_{i,j} = 1 \ (i = 1, 2, \ldots m + n) \tag{58}$$

Here, $A$ represents the connectivity relationship between nodes, if $A_{i,j} = 1$, there is connection between block $P$'s node $i$ and block $Q$'s node $j$, if $A_{i,j} = 0$, there is no connection between block $P$'s node $i$ and block $Q$'s node $j$. $\sum_{i=1}^{m+n} A_{i,j} = 1 \ (j = 1, 2, \ldots m + n)$ and $\sum_{j=1}^{m+n} A_{i,j} = 1 \ (i = 1, 2, \ldots m + n)$ restrict each node to connect to one other node at most. With the solved matrix $A$, the neurite skeletons of adjacent blocks can be merged and fused skeleton structures can be obtained.

## Statistical analysis

In this study, three commonly used metrics defined in *Quan et al., 2016* were used, including precision, recall, and f1-score, which are computed to measure the fidelity between the reconstruction results and the ground truth. They are defined as follows:

$$precision \ (R, G) = \frac{|R \cap G|}{|R|} = \frac{|TP|}{|R|} \tag{59}$$

$$recall \ (R, G) = \frac{|R \cap G|}{|G|} = \frac{|TP|}{|G|} \tag{60}$$

$$f1 - score \ (R, G) = 2 \cdot \frac{precision \times recall}{precision + recall} \tag{61}$$

$R$ represents the point set of reconstructed neurons, $G$ represents the point set of the ground truth, $|\cdot|$ represents the number of points of a set. The three metrics are first computed on each individual neuron and then averaged by weighting each neuron with its point number of its ground truth neuritis.

We also calculated the signal-to-ratio (SNR) of the data using the following method: For a given data block $B$ and its corresponding ground-truth skeleton $S$, we first densify the skeleton $S$ by using linear interpolation to ensure that the Euclidean distance between adjacent skeleton points is less than 1 voxel. Next, we expand each skeleton point in the densified skeleton $S^{'}$ into a spherical mask with a radius of 3 voxels. The resulting region serves as the foreground $mask$. Finally, SNR is computed with mean intensity of foreground points and standard deviation of background points as follows:

$$Mean_{foreground} = \sum_{x \in B} I(x) \times \sigma_1(x) / \sum_{x \in B} \sigma_1(x) \tag{62}$$

$$Mean_{background} = \sum_{x \in B} I(x) \times \sigma_2(x) / \sum_{x \in B} \sigma_2(x) \tag{63}$$

$$Std_{background} = \sqrt{\sum_{x \in B} (I(x) - Mean_{background})^2 \times \sigma_2(x) / \sum_{x \in B} \sigma_2(x)} \tag{64}$$

$$\sigma_1(x) = \begin{cases} 1 & if \left(x \in S^{'}\right) \\ 0 & if \left(x \notin S^{'}\right) \end{cases} \tag{65}$$

$$\sigma_2(x) = \begin{cases} 1 & if \left(x \notin mask\right) \\ 0 & if \left(x \in mask\right) \end{cases} \tag{66}$$

Here, $I(x)$ represents the signal intensity of the voxel at position $x$, the SNR is calculated by $Mean_{foreground}$ and $Std_{background}$ by the following formula:

$$SNR = 10 log_{10} \left(Mean_{foreground} / Std_{background}\right) \tag{67}$$

## Acknowledgements

We thank the members of the Britton Chance Center for Biomedical Photonics for advice and help in experiments. This work was supported by the National Natural Science Foundation of China (32471146) and the project N20240194.

## Additional information

### Funding

| Funder | Grant reference number | Author |
| --- | --- | --- |
| National Natural Science Foundation of China | 32471146 | Tingwei Quan |
| PLA General Hospital | N20240194 | Tingwei Quan |

The funders had no role in study design, data collection and interpretation, or the decision to submit the work for publication.

### Author contributions

Lin Cai, Validation, Investigation, Methodology, Writing – original draft, Writing – review and editing; Taiyu Fan, Validation, Investigation, Writing – original draft; Xuzhong Qu, Software, Investigation; Ying Zhang, Quanwei Ding, Tingting Cao, Validation; Xianyu Gou, Weihua Feng, Investigation; Xiaohua Lv, Xiuli Liu, Qing Huang, Shaoqun Zeng, Supervision; Tingwei Quan, Supervision, Funding acquisition, Investigation, Methodology, Writing – original draft, Writing – review and editing

### Author ORCIDs

Lin Cai ⓘ https://orcid.org/0000-0002-4413-3599
Tingwei Quan ⓘ https://orcid.org/0000-0002-8393-4292

### Ethics

All animal experiments followed procedures approved by the Institutional Animal Ethics Committee of the Huazhong University of Science and Technology.

Reviewer #1 (Public review): https://doi.org/10.7554/eLife.102840.3.sa1
Reviewer #2 (Public review): https://doi.org/10.7554/eLife.102840.3.sa2
Author response https://doi.org/10.7554/eLife.102840.3.sa3

## Additional files

### Supplementary files

MDAR checklist

### Data availability

The data for Figure 1C, Figure 2, Figure 3, Figure 4, Figure 5, Figure 6 and Figure 6—figure supplement 2 is available in https://zenodo.org/records/15589145. The training code of the segmentation network is available on Github: https://github.com/FateUBW0227/Seg_Net (copy archived at *Cai, 2025*). The software of PointTree and its user guideline are available at https://zenodo.org/records/15589145.

The following dataset was generated:

| Author(s) | Year | Dataset title | Dataset URL | Database and Identifier |
| --- | --- | --- | --- | --- |
| Lin C | 2025 | Dataset for PointTree: Automatic and accurate reconstruction of long-range axonal projections of single-neuron | https://doi.org/10.5281/zenodo.15589145 | Zenodo, 10.5281/zenodo.15589145 |

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
