## [Editor Report · eLife Assessment]

This **important** paper takes a novel approach to the problem of automatically reconstructing long-range axonal projections from stacks of images. The key innovation is to separate the identification of sections of an axon from the statistical rules used to constrain global structure. The authors provide **compelling** evidence that their method is a significant improvement over existing measures in circumstances where the labelling of axons and dendrites is relatively dense.

---

## [Referee Report · Reviewer #1 (Public review)]

Summary:

The authors introduce a novel algorithm for the automatic identification of long-range axonal projections. This is an important problem as modern high-throughput imaging techniques can produce large amounts of raw data, but identifying neuronal morphologies and connectivities requires large amounts of manual work. The algorithm works by first identifying points in three-dimensional space corresponding to parts of labelled neural projections, these are then used to identify short sections of axon using an optimisation algorithm and the prior knowledge that axonal diameters are relatively constant. Finally, a statistical model that assumes axons tend to be smooth is used to connect the sections together into complete and distinct neural trees. The authors demonstrate that their algorithm is far superior to existing techniques, especially when a dense labelling of the tissue means that neighbouring neurites interfere with the reconstruction. Despite this improvement, however, the accuracy of reconstruction remains below 90%, so manual proof-reading is still necessary to produce accurate reconstructions of axons.

Strengths:

The new algorithm combines local and global information to make a significant improvement on the state-of -the-art for automatic axonal reconstruction. The method could be applied more broadly and might have applications to reconstructions of electron microscopy data, where similar issues of high-throughput imaging and relatively slow or inaccurate reconstruction remain.

Weaknesses:

There are three weaknesses with the algorithm and manuscript.

(1) The best reconstruction accuracy is below 90%, which does not fully solve the problem of needing manual proof-reading.

(2) The 'minimum information flow tree' model the authors use to construct connected axonal trees has the potential to bias data collection. In particular, the assumption that axons should always be as smooth as possible is not always correct. This is a good rule-of-thumb for reconstructions, but real axons in many systems can take quite sharp turns and this is also seen in the data presented in the paper (Fig 1C). I would like to see explicit acknowledgement of this bias in the current manuscript and ideally a relaxation of this rule in any later versions of the algorithm.

(3) The writing of the manuscript is not always as clear as it could be. The manuscript would benefit from careful copy editing for language, and the Methods section in particular should be expanded to more clearly explain what each algorithm is doing. The pseudo code of the Supplemental Information could be brought into the Methods if possible as these algorithms are so fundamental to the manuscript.

Comments on revisions: I have no further comments or recommendations.

---

## [Referee Report · Reviewer #2 (Public review)]

The authors have addressed my comments in this revised version of their manuscript. PointTree is an improved method for the reconstruction of neuronal anatomy that will be useful for neuroscientists.

In this manuscript, Cai et al. introduce PointTree, a new automated method for the reconstruction of complex neuronal projections. This method has the potential to drastically speed up the process of reconstructing complex neurites. The authors use semi-automated manual reconstruction of neurons and neurites to provide a 'ground-truth' for comparison between PointTree and other automated reconstruction methods. The reconstruction performance is evaluated for precision, recall and F1-score and positions. The performance of PointTree compared to other automated reconstruction methods is impressive based on these 3 criteria.

As an experimentalist, I will not comment on the computational aspects of the manuscript. Rather, I am interested in how PointTree's performance decrease in noisy samples. This is because many imaging datasets contain some level of background noise for which the human eye appears essential for accurate reconstruction of neurites. Although the samples presented in Figure 5 represent an inherent challenge for any reconstruction method, the signal to noise ratio is extremely high (also the case in all raw data images in the paper). It would be interesting to see how PointTree's performance change in increasingly noisy samples, and for the author to provide general guidance to the scientific community as to what samples might not be accurately reconstructed with PointTree.

---

## [Author Response]

The following is the authors’ response to the original reviews

**Public Reviews:**

**Reviewer #1 (Public review):**
Summary:The authors introduce a novel algorithm for the automatic identification of longrange axonal projections. This is an important problem as modern high-throughput imaging techniques can produce large amounts of raw data, but identifying neuronal morphologies and connectivities requires large amounts of manual work. The algorithm works by first identifying points in three-dimensional space corresponding to parts of labelled neural projections, these are then used to identify short sections of axons using an optimisation algorithm and the prior knowledge that axonal diameters are relatively constant. Finally, a statistical model that assumes axons tend to be smooth is used to connect the sections together into complete and distinct neural trees. The authors demonstrate that their algorithm is far superior to existing techniques, especially when dense labelling of the tissue means that neighbouring neurites interfere with the reconstruction. Despite this improvement, however, the accuracy of reconstruction remains below 90%, so manual proofreading is still necessary to produce accurate reconstructions of axons.Strengths:The new algorithm combines local and global information to make a significant improvement on the state-of-the-art for automatic axonal reconstruction. The method could be applied more broadly and might have applications to reconstructions of electron microscopy data, where similar issues of highthroughput imaging and relatively slow or inaccurate reconstruction remain.

We thank the reviewer for their positive comments and for taking the time to review our manuscript. We are truly grateful that the reviewer recognized the value of our method in automatically reconstructing long-range axonal projections. While we report that our method achieves reconstruction accuracy of approximately 85%, we fully acknowledge that manual proofreading is still necessary to ensure accuracy greater than 95%. We also appreciate the reviewer’s insightful suggestion regarding the potential adaptation of our algorithm for reconstructing electron microscopy (EM) data, where similar challenges in high-throughput imaging and relatively slow or inaccurate reconstruction persist. We look forward to exploring ways to integrate our method with EM data in future work.

Weaknesses:There are three weaknesses in the algorithm and manuscript.(1) The best reconstruction accuracy is below 90%, which does not fully solve the problem of needing manual proofreading.

We sincerely appreciate the reviewer's valuable insights regarding reconstruction accuracy. Indeed, as illustrated in Figure S4, our current best automated reconstruction accuracy on fMOST data is still below 90%. This indicates that manual proofreading remains essential to ensure reliability.

For the reconstruction of long-range axonal projections, ensuring the accuracy of the reconstruction process necessitates manual revision of the automatically generated results. Existing literature has demonstrated that a higher accuracy in automatic reconstruction correlates with a reduced need for manual revisions, thereby facilitating an accelerated reconstruction process (Winnubst et al., Cell 2019; Liu et al., Nature Methods 2025).

As the reviewer rightly points out, achieving an accuracy exceeding 95% currently necessitates manual proofreading. Although our method does not completely eliminate this requirement, it significantly alleviates the proofreading workload by: (1) Minimizing common errors in regions with dense neuron distributions; (2) Providing more reliable initial reconstructions; and (3) Reducing the number of corrections needed during the proofreading process.

In the future, we will continue to enhance our reconstruction framework. As imaging systems achieve higher signal-to-noise ratios and deep learning techniques facilitate more accurate foreground detection, we anticipate that our method will attain even greater reconstruction accuracy. Furthermore, we plan to develop a software system capable of predicting potential error locations in our automated reconstruction results, thereby streamlining manual revisions. This approach distinguishes itself from existing models by obviating the need for individual traversal of the brain regions associated with each neuron reconstruction.

(2) The 'minimum information flow tree' model the authors use to construct connected axonal trees has the potential to bias data collection. In particular, the assumption that axons should always be as smooth as possible is not always correct. This is a good rule-of-thumb for reconstructions, but real axons in many systems can take quite sharp turns and this is also seen in the data presented in the paper (Figure 1C). I would like to see explicit acknowledgement of this bias in the current manuscript and ideally a relaxation of this rule in any later versions of the algorithm.

We appreciate the reviewer's insightful opinion regarding the potential bias introduced by our minimum information flow tree model. The reviewer is absolutely correct in noting that while axon smoothness serves as a useful reconstruction heuristic, it should not be treated as an absolute constraint given that real axons can exhibit sharp turns (as shown in Figure 1C). In response to this valuable feedback, we add explicit discussion of this limitation in Discussion section as follow: “Finally, the minimal information flow tree’s fundamental assumption, that axons should be as smooth as possible does not always hold true.

In fact, real axons can take quite sharp turns leading the algorithm to erroneously separate a single continuous axon into disjoint neurites.”

In our reconstruction process, the post-processing approach partially mitigates erroneous reconstructions derived from this rule. Specifically: The minimum information flow tree will decompose such structures into two separate branches (Fig. S7A), but the decomposition node is explicitly recorded. The newly decomposed branches attempt to reconnect by searching for plausible neurites starting from their head nodes (determined by the minimum information flow tree). If no connectable neurites are found, the branch is automatically reconnected to its originally recorded decomposition node (Fig. S7B). In Fig.S7C, two reconstruction examples demonstrate the effectiveness of the post-processing approach.

As pointed out by the reviewers, the proposed rule for revising neuron reconstruction does not encompass all scenarios. Relaxing the constraints of this rule may lead to numerous new erroneous connections. Currently, the proposed rule is solely based on the positions of neurite centerlines and does not integrate information regarding the intensity of the original images or segmentation data. Incorporating these elements into the rule could potentially reduce reconstruction errors.

(3) The writing of the manuscript is not always as clear as it could be. The manuscript would benefit from careful copy editing for language, and the Methods section in particular should be expanded to more clearly explain what each algorithm is doing. The pseudo-code of the Supplemental Information could be brought into the Methods if possible as these algorithms are so fundamental to the manuscript.

We sincerely thank the reviewer for these valuable suggestions to improve our manuscript’s clarity and methodological presentation. We have implemented the following revisions:

(1) Language Enhancement: we have conducted rigorous internal linguistic reviews to address grammatical inaccuracies and improve textual clarity.

(2) Methods Expansion and Pseudo-code Integration: we have incorporated all relevant derivations from the Supplementary Materials into the Methods section, with additional explanatory text to clarify the purpose and implementation of each algorithm. All mathematical formulations have been systematically rederived with modifications to variable nomenclature, subscript/superscript notations and identified errors in the original submission. All pseudocode from Supplementary Materials has been integrated into their corresponding methods subsection.

**Reviewer #2 (Public review):**
In this manuscript, Cai et al. introduce PointTree, a new automated method for the reconstruction of complex neuronal projections. This method has the potential to drastically speed up the process of reconstructing complex neurites. The authors use semi-automated manual reconstruction of neurons and neurites to provide a 'ground-truth' for comparison between PointTree and other automated reconstruction methods. The reconstruction performance is evaluated for precision, recall, and F1-score and positions. The performance of PointTree compared to other automated reconstruction methods is impressive based on these 3 criteria.As an experimentalist, I will not comment on the computational aspects of the manuscript. Rather, I am interested in how PointTree's performance decreases in noisy samples. This is because many imaging datasets contain some level of background noise for which the human eye appears essential for the accurate reconstruction of neurites. Although the samples presented in Figure 5 represent an inherent challenge for any reconstruction method, the signal-to-noise ratio is extremely high (also the case in all raw data images in the paper). It would be interesting to see how PointTree's performance changes in increasingly noisy samples, and for the author to provide general guidance to the scientific community as to what samples might not be accurately reconstructed with PointTree.

We thank the reviewer for her/his time reviewing our manuscript and the interest on how PointTree perform on noisy samples. It is important to clarify that PointTree is solely responsible for the reconstruction of neurons from the foreground regions of neural images. The foreground regions of these neuronal images are obtained through a deep learning segmentation network. In cases where the image has a low signal-to-noise ratio, if the segmentation network can accurately identify the foreground areas, then PointTree will be able to accurately reconstruct neurons. In fact, existing deep learning networks have demonstrated their capability to effectively extract foreground regions from low signal-to-noise ratio images; therefore, PointTree is well-suited for processing neuronal images characterized by low signal-to-noise ratios.

In the revised manuscript, we conducted experiments on datasets with varying signal-to-noise ratios (SNR). The results demonstrate that Unet3D is capable of identifying the foreground regions in low-SNR images, thereby supporting the assertion that PointTree has broad applicability across diverse neuronal imaging datasets.

**Recommendations for the authors:**

**Reviewer #2 (Recommendations for the authors):**
It would be interesting to see how PointTree's performance changes in increasingly noisy samples, and for the author to provide general guidance to the scientific community as to what samples might not be accurately reconstructed with PointTree.

We extend our heartfelt gratitude to the reviewer for their insightful suggestion concerning experiments involving different noisy samples. Here are the details of the datasets used:

LSM dataset: Mean SNR = 5.01, with 25 samples, and a volume size of 192×192×192.

fMOST dataset: Mean SNR = 8.68, with 25 samples, and a volume size of 192×192×192.

HD-fMOST dataset: Mean SNR = 11.4, with 25 samples, and a volume size of 192×192×192.

The experimental results reveal that, thanks to the deep learning network's robust feature extraction capabilities, even when working with low-SNR data (as depicted in Figure 4B, first two columns of the top row), satisfactory segmentation results (Figure 4B, first two columns of the third row) were achieved. These results laid a solid foundation for subsequent accurate reconstruction.

PointTree demonstrated consistent mean F1-scores of 91.0%, 90.0%, and 93.3% across the three datasets, respectively. This underscores its reconstruction robustness under varying SNR conditions when supported by the segmentation network. For more in-depth information, please refer to the manuscript section titled "Reconstruction of data with different signal-to-noise ratios" and Figure 4.